# Evaluation of a mobile-based remote aftercare application in cochlear implant users

Xabier Altuna[1]*, Zuriñe Martínez Basterra[1], Maria Montserrat Soriano-Reixach[1], Leyre Andrés Ustárroz[2], Serafín Sánchez-Gómez[2]

**1** Department of Otorhinolaryngology, Donostia University Hospital, Donostia-San Sebastián, Spain,
**2** Department of Otolaryngology, Virgen Macarena University Hospital, Seville, Spain

\* xaltuna@osakidetza.net

## Abstract

### Introduction

Clinical follow-up for cochlear implant (CI) users is typically carried out in a face-to-face session in a clinical site. Users must travel to the site, which incurs burdens of travel time, expenses, and time lost from work or education. To help alleviate this, a mobile health application, Remote Check, has been developed to facilitate the remote evaluation of both the functionality of the CI system and the hearing performance of the user. Here, we report the results of a real-world field test of Remote Check with a cohort of 29 CI users across two clinics.

### Methods

Paired Remote Check and face-to-face clinical follow-up session were performed with a cohort of 29 experienced CI users. A psychometric instrument for the evaluation of mobile health applications (the mHealth App Usability Questionnaire, MAUQ) was administered to both users and clinical professionals, along with an ad-hoc survey to evaluate users' experiences regarding travel distances, travel times, costs, and other parameters associated with different types of clinical follow-up sessions. Professionals were also surveyed on the performance of the tool. The durations of Remote Check and face-to-face follow up sessions were compared, and the agreement rate of post-session follow-up actions was determined.

### Results

Both technical checks of the CI and evaluation of hearing performance were possible with Remote Check. Both users and professionals rated the tool highly on the MAUQ instrument. This tool facilitated shorter session times and had strong agreement with face-to-face follow up in terms of further actions. Users also perceived Remote Check as offering the possibility of saving them money.

**Data availability statement:** All relevant data are within the manuscript and its Supporting information files.

**Funding:** The author(s) received no specific funding for this work.

**Competing interests:** The authors have declared that no competing interests exist.

**Abbreviations:** CI, cochlear implant; MAUQ, mHealth App Usability Questionnaire; PROM, patient reported outcome measure

## Conclusion

MED-EL Remote Check is an adequate tool for the remote evaluation of both the CI system and the users' hearing performance. Remote Check may reduce barriers to care, particularly for rural populations.

## Introduction

Cochlear implants (CI) are the gold standard treatment option for severe-to-profound sensorineural hearing loss. A CI is typically activated within four weeks of the surgical implantation [1]. During this session, a first fitting is conducted, in which the device parameters of the audio processor are optimized for the individual user. This is carried out through both objective hearing perception tests and subjective feedback from the user [2].

After implant activation, regular follow-up sessions take place. In these sessions, the clinical professional provides consultation and conducts a battery of tests to monitor hearing performance and device functionality. Audio processor parameters may also be fine-tuned, if necessary, to account for the development of the users' auditory perception with continued device usage as well as to account for progression of hearing loss when a progressive form is present. These follow-up sessions take place within the context of the wider hearing re-habilitation program, in which speech and auditory training are carried out.

Follow-up sessions typically entail a visit by the user to a specialist at an audiology center. This can pose several challenges and inconveniences. Frequently the user must take time from work or education. Follow-up sessions may entail out-of-pocket expenses for the user [3,4]. For individuals who live far from an audiology center, attending a follow-up session involves the time and expense of transportation and potentially accommodation [5,6]. This is an ongoing constraint; users who frequently travel or visit remote regions must be mindful that they may not have ready access to routine or emergency audiological services. One study found that 88% of adult CI users would prefer remote chare for at least some aspects of their follow-up [7].

For early post-implantation aftercare, in-person visits to the audiology and rehabilitation clinic are probably essential. There may be more scope for employing remote care during later routine follow-up sessions, when the user is already accustomed to hearing with the device, and to their caregivers and the procedures. Ideally, routine follow-up sessions could take place remotely with the consultation and speech perception testing carried while the CI user was at home. Only if significant problems are discovered by the clinical professional or reported by the user, an in-person visit to the clinic is necessary. As a step toward the implementation of fully remote clinical follow-up for CI users, the CI manufacturer MED-EL has developed MED-EL Remote Check. MED-EL Remote Check is a web-based application designed to facilitate remote assessment of aided hearing performance and to collect patient reported outcome measures (PROMs) implemented as questionnaires. It is intended for users of all ages and with any degree of CI experience, (although younger children may

require the assistance of adults). This application is an optional accessory which is compatible with all currently available MED-EL audio processors. MED-EL Remote Check was designed according to a mobile-optimized design, facilitating its use in any location, at home or otherwise. Other applications for remote assessment of CI users have been developed (for example [8–10]). To the best of our knowledge, the present system is the first to incorporate PROM assessments.

Therefore, the primary objective of this study was to compare clinical follow-up sessions using the MED-EL Remote Check application with conventional face-to-face follow-up sessions. The secondary objectives were: 1) to collect data on the usability and user friendliness of the platform from both CI users (and their caregivers) and from clinicians; 2) to collect data on follow-up visits, including the time and costs associated with these; and 3) to compare the diagnoses made through MED-EL Remote Check with those made through face-to-face follow-up sessions.

## Materials and methods

### Study design

This study was designed as a multi-center, open label, one-arm, prospective study. Data collection took place between September 2022 and October 2023.

Both Donostia hospital and Virgen Macarena hospital are tertiary care centers. According to the distribution described in [11], the Donostia hospital falls within the medium/average cochlear implant volume range, whereas the Virgen Macarena corresponds to a large-volume category.

### Ethics and informed consent

This study was designed and conducted in accordance with the Declaration of Helsinki. Ethical approval was provided by the Research Ethics Committee of the Integrated Health Organization (OSI) Donostialdea (approval code: ALT-REM-2022-01) and the Ethics Committee of the University Hospital Virgen Macarena, Seville, Spain (approval code: 0975-N-23). All participants provided their signed and dated informed consent prior to the start of any study-specific procedures. Consent included the publication of these findings. For children (<12 years), informed consent was provided by their caretakers.

### Inclusion and exclusion criteria

Participants were included in this study if there were users of MED-EL CI systems (COMBI40+ or newer). Participants were required to use an audio processor that was compatible with the MED-EL Remote Check system (OPUS 2, SONNET 1/2, or RONDO 1/2/3, all from MED-EL, Innsbruck, Austria) and which can receive input using either a direct input cable (all audio processors), the AudioStream system (SONNET 1/2), the AudioLink system (SONNET1/2 and RONDO 3), or the Artone neckloop (Artone Communication Solutions, New York City, NY). The self-reported ability to operate a mobile device was also a requirement for the patient or their caregiver.

Aside from lack of conformance with the inclusion criteria, participants were excluded from this study if they were implanted with the COMBI40X or COMBI40C CI system, or with an auditory brainstem implant, or a split electrode array (although there were no exclusions made on the basis of these device criteria). We also allowed the exclusion of anything that, in the opinion of the investigators, would place the subject at increased risk or preclude the subject's full compliance with or completion of the study (for example psychological disorders, or neurologic or motoric comorbidities that could impede subjects from using the application).

### Participants

A total of 29 CI users (14 female) participated in this study. We found that 13 were children or adolescents (<18 years) and 16 were adults. Out of the 29 participants, 13 were unilateral CI users, nine were bilateral CI users,

 

and seven were bimodal (CI + hearing aid) users. The majority of adult users were post-lingual (12/16), while a minority of pediatric users were post-lingual (3/13). The median time since first fitting was 8.5 years (range: 1.1–22.6 years). 22 users (76%) wore their audio processor for > 12 hours per day and 7 (24%) wore it for < 12 hours per day.

Three professionals participated in this study: one audiologist with <5 years of experience, one ENT with >10 years of experience, and one speech and language therapist (SLT) with 5–10 years of experience. Two of these reported engaging in > 40 follow-up appointments per month and one reported 10–20 follow-up appointments per month.

## MED-EL remote check

MED-EL Remote Check is a web application which facilitates the remote assessment of hearing and audio processor performance through both objective hearing tests and PROMs implemented as questionnaires.

The application implements several hearing tests. These are a Two-Tone Test (to determine the hearing sensitivity to certain frequencies) [12]; a Three-Digit Test (to assess speech-in-noise perception) [13]; and a Six-Sound Test (comprised of six speech sounds spanning the range of frequencies in spoken language) [14]. The application also implements two questionnaires, one of which assesses the user's hearing in everyday listening situations and one of which assesses the performance of the user's audio processor. All sound perception test material is delivered via direct connection to the audio processor.

For this study, all sessions were conducted in person at the investigating clinics. A MED-EL Remote Check session is first initiated by the clinical professional, who then invites the user to a hearing assessment. The user, after accepting the invitation, completes the hearing tests and the questionnaires with their personal mobile phone. This was carried out either by the user themselves or by their caregiver or companion. The results are then transmitted to the clinical professional. Based on the results, the clinical professional then is able to make a diagnosis and advises the user on the next steps.

## Face-to-face follow-up fittings

To provide a comparison for the MED-EL Remote Check sessions, a conventional face-to-face follow-up session was scheduled with each participant. These sessions were scheduled on the same day, with the Remote Check session being performed first in all cases. These sessions were carried out by the clinical professionals according to their routine protocol at their institution.

## User assessments

**mHealth app usability questionnaire.** To evaluate the usability of MED-EL Remote Check, we administered the mHealth App Usability Questionnaire (MAUQ) [15], which is an instrument designed to assess the usability of mobile health applications. The MAUQ was designed and published in the English language. To use it here, we performed an *ad hoc* translation of the items into Spanish using a forward and back-translation method. A validated Spanish MAUQ variant has since been published [16].

The MAUQ has three subscales which assess the constructs of usefulness, ease of use and satisfaction, and interface and satisfaction. The variant for interactive applications contains 21 items and that for standalone applications contains 18 items. A variant was designed for clinical professionals which contain the same items but with slight textual revisions (such as revising *receiving healthcare* to *delivering healthcare*). In this study, the interactive variants of the MAUQ were administered to the participants and the clinical professions, using the respective variant for each. A forward and back-translation approach was used to produce a Spanish translation of the instrument.

 

Items were rated on a seven-point Likert scale ranging from 1 (strongly disagree) to 7 (strongly agree). Unanswered items were replaced with a score of four (neither agreement nor disagreement). Responses were excluded if they had > 50% of items unanswered, either for the instrument as a whole or for any subscale.

**User survey.** Along with the MAUQ, an ad-hoc survey was developed to assess the frequency with which the users have face-to-face follow-up visits with clinical professionals, as well as the travel, time and cost burdens associated with these visits.

### Clinical professional assessments

**MAUQ.** The MAUQ was administered as described above, but using the variant designed for clinical professionals.

**Clinical professional survey.** Alongside the MAUQ, an ad-hoc survey was developed to assess the types of tasks which are performed during a typical follow-up appointment, as well as their responses to the MED-EL Remote Check system (overall satisfaction, ability to make a clinical diagnosis, ability to advise patients on the next steps, and any complications encountered).

### Assessment of procedure equivalence

To assess the equivalence of the MED-EL Remote Check sessions and face-to-face follow-up sessions, we compared the instances in which further action was needed. At the end of each session, the clinical professional documented whether, based on the information gathered in that session, further action was needed (such as fitting adjustment or a further visit to the clinic), and what that further action was.

The outcome measure was rate of concurrence between the recommendation made using the MED-EL Remote Check modality and the recommendation made during the face-to-face session. The total rate of concurrence was calculated, and separate rates of concurrence were also calculated for audiological, technical, and medical actions.

### Assessment time- and cost-effectiveness

To assess the time-effectiveness of MED-EL Remote Check relative to face-to-face follow-up, the duration of each session with each CI user in each modality was documented. The duration spent with each clinical professional (surgeon, physician, acoustician, CI audiologist, rehabilitation specialist, or organizational personnel such as secretaries) was separately documented. The total duration of each session was documented, as was the individual duration for audiological, technical, and medical actions during each session.

To assess the cost-effectiveness of MED-EL Remote Check, the users were asked, as part of the user survey, to rate the degree to which they agreed with the statement *Remote Check allows me to save money*. Agreement with the statement was evaluated on a seven-point Likert-type scale, ranging from 1 (strongly disagree) to 7 (strongly agree).

### Statistical analysis

Descriptive statistics were calculated to report participants' baseline characteristics and to describe test outcomes. The mean, standard deviation (SD) and/or median with range were used to describe quantitative data. Absolute and relative frequencies were used to describe qualitative data.

## Results

### MAUQ responses

In the current study, 20 users fully completed the MAUQ, while two additional users only partially completed the questionnaire (Table 1). Similar scores were reported for the three domains of *ease of use*, *system information arrangement*, and *usefulness*. The mean total score for the MAUQ was 5.8 points (±1.0 point SD). The mean scores on the domains were 5.9 (±1.0) for *ease of use*, 5.8 (±0.9) for *system information arrangement*, and 5.8 (±1.2) for *usefulness*.

**Table 1. User responses on the MAUQ instrument.**

|  | n | Mean | SD | Min | Max |
|---|---|---|---|---|---|
| MAUQ total score | 25 | 5.8 | 1.0 | 3.7 | 7.0 |
| *Ease of use* | 27 | 5.9 | 1.0 | 3.3 | 7.0 |
| *System information arrangement* | 20 | 5.8 | 0.9 | 4.0 | 7.0 |
| *Usefulness* | 25 | 5.8 | 1.2 | 3.1 | 7.0 |

Professional responses on the MAUQ are given in Table 2. Generally, the ENT gave lower ratings compared to the other professionals. The highest ratings were given by the SLT on the *ease of use* and *usefulness* domains (6.3), while the lowest ratings were given by the ENT on the *usefulness* domain.

## User survey

Responses from the user survey are given in Table 3. We found that 28 users completed the survey, although not all users responded to all questions. Most of the users reported that they visited both CI audiologists and ENT specialists/ CI surgeons at least once per year, while the majority reported that they did not visit rehabilitation. Out of the 19 users who did not attend rehabilitation (six of whom were children), nine remarked that they had either been discharged or completed rehabilitation, while seven reported that they did not feel that rehabilitation was necessary.

For visits to both CI audiologists and ENT specialist/ CI surgeons, the most frequently reported distance travelled was > 100 km, while for the minority who attended rehabilitation, the most frequently reported distance was 11–20 km. The most frequently reported travel time was 3–4 h for all types of visits.

Most users reported that the costs of these visits were < 50 €. For visits to both CI audiologists and rehabilitation, support for the visit was frequently required, while the majority reported that support was not needed for visits to ENT specialist/ CI surgeons. A day off from work, school, or university was required by the majority for visits to both CI audiologists and ENT specialists/ CI surgeons, but not for visits to rehabilitation. When a day off was required, the majority did not report that the time off from work was paid.

Additionally, users were asked to rate whether they agreed with the statement *I recommend Remote Check to others* on a seven-point Likert-type scale ranging from 1 (strongly disagree) to 7 (strongly agree). Out of the 24 users who responded, 12 strongly agreed with the statement. Two users disagreed with this statement, rating it either 1 or 2. The remaining ten rated the statement with 4 (n = 1), 5 (n = 1), or 6 (n = 8). Similar responses were provided for both pediatric users (mean 6.20) and adult users (mean 6.38).

## Professional survey

Responses from the professional survey are given in Table 4. All respondents reported using MED-EL Remote Check for technical evaluation of the CI system and for evaluation of hearing performance.

**Table 2. Professional responses on the MAUQ instrument. Individual responses from each of the professionals are shown.**

|  | ENT | Audiologist | SLT | Mean | SD | Median |
|---|---|---|---|---|---|---|
| MAUQ total score | 4.0 | 6.0 | 6.2 | 5.4 | 1.2 | 6.0 |
| *Ease of use* | 5.0 | 6.0 | 6.3 | 5.8 | 0.7 | 6.0 |
| *System information arrangement* | 4.8 | 6.0 | 6.2 | 5.7 | 0.8 | 6.0 |
| *Usefulness* | 2.0 | 6.0 | 6.3 | 4.8 | 2.4 | 6.0 |

**Table 3. Response frequencies from the user survey. Conditional items, based on responses from primary items, are given in italics.**

| | | CI audiologist | | ENT specialist/ CI surgeon | | Rehabilitation | |
|---|---|---|---|---|---|---|---|
| | | n | % | n | % | n | % |
| Visits | Yes | 27 | 96.4 | 20 | 71.4 | 8 | 28.6 |
| | *Multiple times a month* | 0 | 0 | 0 | 0 | 5 | 17.9 |
| | *Every 6 months* | 8 | 28.6 | 3 | 10.7 | 2 | 7.1 |
| | *Once a year* | 18 | 64.3 | 16 | 57.1 | 1 | 3.6 |
| | *Less than once a year* | 1 | 3.6 | 1 | 3.6 | 0 | 0 |
| | No | 1 | 3.6 | 8 | 28.6 | 19 | 67.9 |
| | *I don't feel the necessity* | 1 | 3.6 | 4 | 14.3 | 7 | 25 |
| | *Other reasons* | 0 | 0 | 3 | 10.7 | 10 | 35.7 |
| | Not reported | 0 | 0 | 0 | 0 | 1 | 3.6 |
| Distance travelled | <5 km | 2 | 7.1 | 3 | 10.7 | 0 | 0 |
| | 5–10 km | 4 | 14.3 | 1 | 3.6 | 2 | 7.1 |
| | 11–20 km | 1 | 3.6 | 1 | 3.6 | 3 | 10.7 |
| | 21–40 km | 3 | 10.7 | 2 | 7.1 | 1 | 3.6 |
| | 41–70 km | 5 | 17.9 | 3 | 10.7 | 1 | 3.6 |
| | 71–100 km | 1 | 3.6 | 0 | 0 | 0 | 0 |
| | > 100 km | 11 | 39.3 | 9 | 32.1 | 1 | 3.6 |
| | Not reported | 1 | 3.6 | 9 | 32.1 | 20 | 71.4 |
| Time required | <2h | 9 | 32.1 | 6 | 21.4 | 3 | 10.7 |
| | 3–4 h | 11 | 39.3 | 7 | 25.0 | 5 | 17.9 |
| | 5–6 h | 4 | 14.3 | 3 | 10.7 | 0 | 0 |
| | 7–8 h | 1 | 3.6 | 1 | 3.6 | 0 | 0 |
| | 9–23 h | 2 | 7.1 | 2 | 7.1 | 0 | 0 |
| | Not reported | 1 | 3.6 | 9 | 32.1 | 20 | 71.4 |
| Costs | < 50 € | 18 | 64.3 | 13 | 46.4 | 8 | 28.6 |
| | 51-100 € | 6 | 21.4 | 4 | 14.3 | 0 | 0 |
| | 101-150 € | 1 | 3.6 | 1 | 3.6 | 0 | 0 |
| | > 150 € | 2 | 7.1 | 1 | 3.6 | 0 | 0 |
| | Not reported | 1 | 3.6 | 9 | 32.1 | 20 | 71.4 |
| Support required for visit | Yes | 16 | 57.1 | 8 | 28.6 | 6 | 21.4 |
| | No | 11 | 39.3 | 9 | 32.1 | 2 | 7.1 |
| | Not reported | 1 | 3.6 | 11 | 39.3 | 20 | 71.4 |
| Day off required | Yes | 18 | 64.3 | 11 | 39.3 | 2 | 7.1 |
| | *Paid leave* | 9 | 32.1 | 4 | 14.3 | 1 | 3.6 |
| | *Unpaid leave* | 9 | 32.1 | 7 | 25.0 | 1 | 3.6 |
| | No | 7 | 25.0 | 6 | 21.4 | 5 | 17.9 |
| | Not reported | 3 | 10.7 | 11 | 39.3 | 21 | 75.0 |

All reported that the system enabled the making of a clinical diagnosis and making it possible to advise the patient of the next steps, for both adults and for children/adolescents. One respondent reported complications occurring during the session. Two respondents reported being satisfied or very satisfied with the system.

**Table 4. Response frequencies from the professional survey. RC: MED-EL Remote Check.**

| Category | Task or response | n |
|---|---|---|
| Tasks performed in a typical follow-up session | Technical evaluation of the CI system | 3 |
| | Evaluation of fitting aspects of the CI system | 2 |
| | Evaluation of hearing performance | 3 |
| | Evaluation of speech understanding (speech test in quiet, speech test in noise) | 2 |
| | Medical check-up | 2 |
| RC enabled the making of a clinical diagnosis | Yes | 3 |
| | No | 0 |
| RC made it possible to advise your patients of the next steps | Yes | 3 |
| | No | 0 |
| Any complications encountered | Yes | 1 |
| | No | 2 |
| Overall satisfaction with RC | Very satisfied | 1 |
| | Satisfied | 1 |
| | Neutral | 1 |

## Time- and cost-effectiveness

The time difference in minutes between MED-EL Remote Check sessions and face-to-face follow-up sessions are shown in Table 5. Time differences are stratified according to task (audiological, technical, and medical), as well as the total combined difference. All tasks had a shorter mean duration with MED-EL Remote Check as compared to face-to-face follow-up.

The distribution of user responses (n = 27) to the statement *Remote Check allows me to save money* is shown in Fig 1. Almost half (48%) strongly agreed with this statement. One respondent strongly disagreed with this statement. Similar responses were provided for both pediatric users (mean 5.83) and adult users (mean 6.22).

## Procedure equivalence between MED-EL Remote Check and face-to-face follow-up

The results of the procedure equivalent assessment are summarized in Fig 2. For both technical and medical follow-up actions, both the MED-EL Remote Check sessions and face-to-face follow-up sessions had full agreement. For audiological follow-up actions, 3 of 30 session pairs (10%) did not have agreement.

**Table 5. Time difference in minutes between MED-EL Remote Check sessions and face-to-face follow-up sessions. Negative values indicate shorter times for MED-EL Remote Check sessions. SD: standard deviation.**

| | Mean (SD) | Median (min, max) |
|---|---|---|
| Audiological | −10.8 (5.2) | −9 (0, −19) |
| Technical | −7.6 (7.7) | −8 (2, −28) |
| Medical | −4.1 (4.4) | −3 (0, −18) |
| Combined | −22.5 (10.3) | −20 (−1, −40) |

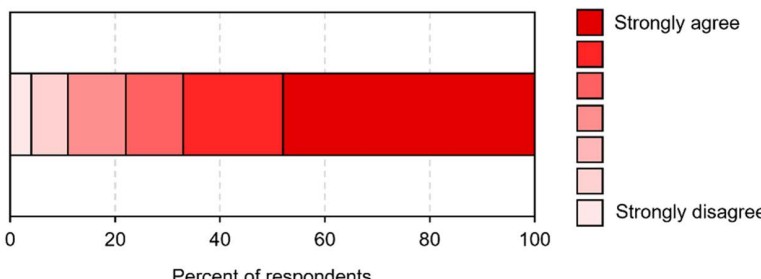

**Fig 1. Distribution of user responses to the statement** *Remote Check allows me to save money.*

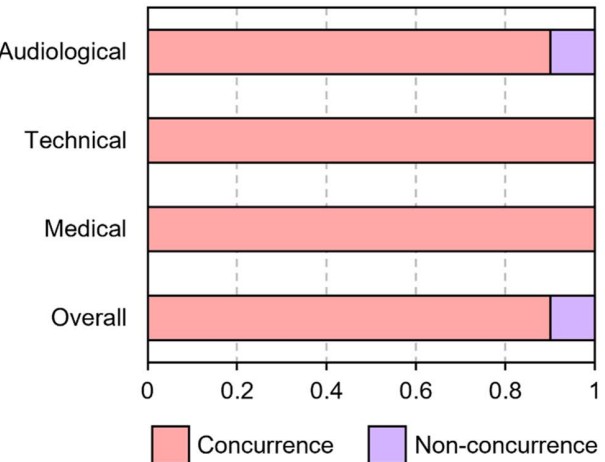

**Fig 2. Ratings of concurrence between MED-EL Remote Check and face-to-face follow-up for audiological, technical, and medical actions.** The overall concurrence rate is also shown.

## Discussion

### Summary of findings

In general, high (favorable) scores were observed on all three domains of the MAUQ for both users (adults and children/adolescents) (mean total score 5.8/7), and professionals (mean total score 5.4/7). The ENT gave somewhat lower scores than the other professionals. It is possible that ENTs prioritize or require in-person examinations. To the best of our knowledge, there are no established methods to categorize the results of the MAUQ (i.e., cutoff values for usability rated as poor, acceptable, good, etc.). Following Sanchez Roman et al. [17], we suggest provisionally aligning MAUQ scores to the categories of the more general System Usability Scale (SUS [18]). This would yield lower cutoff values for the MAUQ of poor (2.7), OK (3.7), good (5.1), and excellent (6.0), and an overall acceptable usability threshold of 4.9.

From the user survey, it was observed that most reported attending both CI audiologists and ENT specialists/ CI surgeons at least once per year. Of these, a substantial number (40–47%) reported that they needed to travel relatively long distances (> 100 km) to attend these appointments. This incurred both long travel times (most frequently 3–4 h, but up to 9–23 h in some cases), and financial burdens including the need to take unpaid time off from work or education. It should be noted that in cases of pediatric users, these burdens may be increased by the fact that a parent or guardian may also need to take a day off from work to support the visit. Thus, in these scenarios, both a day of education and a day

of work are both lost in order to facilitate a visit to the clinic. These results, therefore, justify the approach of using remote application-based support for CI users to supplement face-to-face care, as is proposed with the MED-EL Remote Check application.

From the professional survey, it was found that all respondents were able to use the system for both technical evaluation of the CI and for evaluation of hearing performance. All reported that the system enabled the making of a clinical diagnosis and making it possible to advise the patient of the next steps. One respondent reported complications occurring during one session due to problems with the direct audio input connection. Two respondents reported being satisfied or very satisfied with the system.

It was also found that MED-EL Remote Check sessions had a shorter duration than face-to-face sessions. This was true for all task types evaluated – audiological, technical, and medical. For all tasks combined, MED-EL Remote Check sessions had a median reduction in duration of 20 minutes (minimum: 1 minute, maximum: 40 minutes) relative to face-to-face sessions. Although the variance was quite large, it can nevertheless be stated that MED-EL Remote Check facilitates shorter session times.

To estimate the cost-effectiveness of MED-EL Remote Check sessions, we surveyed users about their degree of agreement with the statement *MED-EL Remote Check allows me to save money.* We found that 77% of respondents gave positive responses to this statement (ratings > 4 on the 7-point Likert scale), and almost half (48%) strongly agreed with this statement. Three users (11%) gave ratings < 4, and one of these strongly disagreed (rating of 1). All of these were adult users. We do not know the reasoning of the individual who strongly disagreed. It is reasonable to suppose that financial savings will be less for those who are not exposed to major costs (for example, those living close to the clinic or those who are retired and will do not need to take time off from work). This individual nevertheless provided relatively high scores on all domains of the MAUQ (*ease of use*: 7.0, *system information arrangement*: 6.33, *usefulness*: 7.0). From these data, it can be observed that most users perceive MED-EL Remote Check as allowing them to reduce the financial burden associated with their clinical follow-up.

To test the procedural equivalence of MED-EL Remote Check and face-to-face sessions, we recorded the agreement rate of follow-up actions for both session types. Full agreement was observed for both technical and medical actions. 3 of 30 session pairs (10%) did not have agreement, all of which were for audiological follow-up actions. It is unclear why full agreement was not achieved with audiological actions. It may be the case that these types of actions require a relatively greater degree of in-person interaction.

In summary, it was found that as a medical healthcare application, MED-EL Remote Check was rated highly by both users and professionals, that it was possible for the professional to both perform technical checks of the CI and to evaluate hearing performance, and that MED-EL Remote Check facilitated shorter session times, and lead to largely the same clinical follow-up actions. Users and/or parents or caregivers of users perceived MED-EL Remote Check as offering the possibility of saving them money. Although not directly measured, it can be inferred that the use of MED-EL Remote Check can also save time for the user by eliminating the need to travel for their follow-up visits. These benefits may be of particular value in cases of pediatric users, who are typically accompanied to follow-up sessions by a parent or guardian, who must also take time off from work, and finance their own travel, to facilitate the appointment.

## Prior studies and other CI telehealth applications

Previous studies have evaluated other mobile health applications to facilitate remote hearing performance testing of CI users. One study demonstrated the usability and motivation for uptake among older-aged CI users of an application (MyHearingApp) designed for self-testing of hearing performance [10]. The results of the present study are broadly in accord with this, in that users found both applications to be relatively highly usable, according to the MAUQ in our study and the SUS in Philips et al.

Another application (Remote Check from Cochlear Ltd.) has also been evaluated for self-testing of hearing performance of both adult and pediatric CI users [8,9,19,20]. This application was found to be easy to use by most CI users in the study, and the self-administered testing was found to be in good agreement with clinic-based testing.

Another mobile application (hearDigits) has been tested for the evaluation of speech reception of adult CI users. This application implemented a digits-in-noise test to determine speech reception thresholds, as well as the Three-digit test that is included in the web app evaluated here. Also, high correlation between clinical testing and testing through the app were reported [21] and reliability was evaluated through test-retest.

The World Health Organization has also produced a mobile application for hearing screening (hearWHO) which also implements the digits-in-noise [22]. This application has been administered to adult listeners in multiple countries and is envisaged to facilitate large-scale hearing screening campaigns. It is available in English, Dutch, Mandarin, Russian and Spanish.

Along with this, remote self-evaluation of implant telemetry has also been studied. One study observed that self-evaluation of electrode impedance levels could be accomplished by users at home, with the tests results being transmitted online for remote monitoring [23]. Remote testing of electrically evoked compound action potentials has also been accomplished [24]. At present, these telemetry functions are not implemented in Remote Check.

### Speech perception tests

In the Remote Check app, speech perception tests are implemented as a three-digit test to asses speech-in-noise perception and the six speech sound perception test to assess access to speech sounds. Ideally, it would be possible to implement a wider set of tests of speech perception, particularly monosyllabic words in quiet and sentences in noise, as are commonly used in clinical testing. There are a few hurdles to this. For one, the test material is generally proprietary, so the app would have to implement custom material which may not have the same psychometric properties as those used in the clinic.

### Study limitations

This study has several limitations which affect both the interpretation and the generalizability of our findings. First, this study has a relatively small sample size. Statistical inference from a cohort of this size is often accompanied by a degree of uncertainty and was, therefore, not attempted. Not all users completed all assessments, for example only 20 users fully completed the MAUQ (response rate 71%). We cannot know if there was a systematic bias such that non-responding users had different opinions. Second, the study was performed at only two clinics, and that the degree to which these findings will generalize to other clinics is unknown. Third, the study was performed in the Spanish language, but proficiency in this language was not added as an inclusion criterion. As we performed an *ad hoc* forward-backward translation of the MAUQ to Spanish, it cannot be certain that the psychometric properties of this instrument (factor loadings, construct and criterion validity, and internal consistency) carry over into the Spanish language.

### Conclusions

Here we have shown that the MED-EL Remote Check application can be used to facilitate remote evaluation of both the CI system and of the users' hearing performance. Both users and professionals found the application to be usable. It is suitable for use by both adult and pediatric users. Good agreement was observed between clinical follow-up actions after sessions with Remote Check and conventional face-to-face session.

### Supporting information

**S1 File. Remote Check complete info anon.**
(PDF)

## Acknowledgments

The authors would like to express their gratitude to all of the individuals who participated in this study. The authors would also like to thank Elena Muñoz Pascual, Maria del Mar Barrios Romero, Mareike Billinger-Finke, and Patrick Connolly of MED-EL for assistance with various aspects of this manuscript.

## Author contributions

**Investigation:** Xabier Altuna, Maria Montserrat Soriano-Reixach, Leyre Andrés Ustárroz, Serafin Sanchez-Gómez.

**Methodology:** Xabier Altuna, Zuriñe Martínez Basterra, Maria Montserrat Soriano-Reixach, Serafin Sanchez-Gómez.

**Supervision:** Xabier Altuna, Maria Montserrat Soriano-Reixach.

**Writing – original draft:** Xabier Altuna.

**Writing – review & editing:** Xabier Altuna.

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
