## [Decision Letter · Decision Letter 0]

19 Nov 2025

Dear Dr. ALTUNA,

Thank you for submitting your manuscript to PLOS ONE. After careful consideration, we feel that it has merit but does not fully meet PLOS ONE’s publication criteria as it currently stands. Therefore, we invite you to submit a revised version of the manuscript that addresses the points raised during the review process.

We look forward to receiving your revised manuscript.

Kind regards,

Rohit Ravi, Ph.D.

Academic Editor

PLOS ONE

Journal Requirements:

3. We note that your Data Availability Statement is currently as follows: All relevant data are within the manuscript and in Supporting Information files.

Additional Editor Comments:

I have assessed the comments of both the reviewers, they recommend revising the manuscript, make sure you address the comments raised by both reviewers.

Reviewers' comments:

Reviewer's Responses to Questions

**Comments to the Author**

1. Is the manuscript technically sound, and do the data support the conclusions?

Reviewer #1: Yes

Reviewer #2: Partly

2. Has the statistical analysis been performed appropriately and rigorously?

Reviewer #1: Yes

Reviewer #2: No

3. Have the authors made all data underlying the findings in their manuscript fully available?

Reviewer #1: Yes

Reviewer #2: Yes

4. Is the manuscript presented in an intelligible fashion and written in standard English?

Reviewer #1: Yes

Reviewer #2: Yes

Reviewer #1: Your manuscript is well-structured. The study effectively compares remote vs. in-person follow-up, addressing a significant gap in CI care. The use of validated tools (MAUQ) and ad-hoc surveys adds rigor. The focus on cost/time savings for patients and clinics is timely and impactful. Strengths include robust usability metrics and clear clinical relevance. The manuscript is publishable with minor revisions. Below is a detailed review with suggestions for improvement, organized by section:

1. Abstract

• Clarity: The abstract succinctly summarizes key findings but could briefly mention the sample size (n=29) earlier for context.

• Conclusion: Consider adding a line about the clinical implications (e.g., "Remote Check may reduce barriers to care, particularly for rural populations").

2. Introduction

• Flow: The transition from CI follow-up challenges to Remote Check’s development is smooth.

• Gap: Explicitly state how this study advances prior work (e.g., "While other apps exist [cite Philips et al., Maruthurkkara et al.], Remote Check uniquely combines technical CI checks with hearing performance tests").

3. Methods

• Clarify if consent included data publication (if applicable).

• Inclusion/Exclusion:

o Specify why COMBI40X/C users were excluded (technical incompatibility?).

o Clarify how "ability to operate a mobile device" was assessed (self-reported? screening test?).

• MAUQ Translation: Briefly note if the translation was validated (or cite a reference for the method).

4. Results

• MAUQ Scores:

o The high scores (mean ~5.8/7) are compelling, but consider adding a comparison to benchmarks (e.g., "Scores exceed the ‘acceptable’ threshold of 4.5/7 in mHealth studies [cite]").

o Address the ENT’s lower ratings in the Discussion (e.g., "ENTs may prioritize in-person exams for complex cases").

• Cost Savings:

o The 48% "strongly agree" is impactful, but contextualize the one dissenting adult (e.g., "The outlier may reflect unique financial circumstances").

• Procedure Agreement:

o Highlight the 10% audiological disagreement rate as a limitation (e.g., "Audiological actions may require supplemental in-person checks").

5. Discussion

• Limitations:

o Sample Size: Acknowledge potential Type II errors (e.g., "Small n may underdetect differences in subgroups like pediatric vs. adult users").

o Generalizability: Note if clinics were urban/rural or had high/low CI volumes.

• Comparison to Prior Work:

o Emphasize how Remote Check differs (e.g., "Unlike hearDigits, Remote Check integrates PROMs and technical checks").

• Clinical Implications:

o Recommend hybrid models (e.g., "Remote Check for routine checks; in-person for audiological fine-tuning").

6. Language & Clarity

• Minor Edits:

o "Mobile-first principles" → "Mobile-optimized design" (more precise).

o "Ad-hoc survey" → "Custom survey" (avoids informal tone).

• Repetition: Some results (e.g., MAUQ scores) are duplicated; streamline.

Reviewer #2: The present study objective is to compare clinical follow-up sessions using the MEDEL remote check application with conventional face-to-face follow-up sessions. The specific domains are usability and user friendliness of the platform from CI users/caregivers/clinicians, time and costs involved, and comparing the diagnosis made through two modalities (remote control and face-to-face sessions). I appreciate the authors for making an effort to assess the use of a remote check web-based application in the MEDEL cochlear device. There are section-specific comments as an attachment to be addressed and looked into for strengthening the manuscript.

**Do you want your identity to be public for this peer review?** For information about this choice, including consent withdrawal, please see our Privacy Policy

Reviewer #1: **Yes:** Mohammed Elrabie Ahmed

Reviewer #2: No

---

## [Author Response · Author response to Decision Letter 1]

6 Jan 2026

Dear Editor and reviewers,

We sincerely thank you for your careful review and comments, which will improve the quality of our manuscript.

Please find attached the revised manuscript for the study We thank the editor and reviewers for their thorough input. We hope the revised version satisfies the requested revisions.

All changes have been marked up with green highlight.

Reviewer #1

1. Abstract

• Clarity: The abstract succinctly summarizes key findings but could briefly mention the sample size (n=29) earlier for context.

Thanks, added.

• Conclusion: Consider adding a line about the clinical implications (e.g., "Remote Check may reduce barriers to care, particularly for rural populations").

Likewise, we have added this.

2. Introduction

• Flow: The transition from CI follow-up challenges to Remote Check’s development is smooth.

• Gap: Explicitly state how this study advances prior work (e.g., "While other apps exist [cite Philips et al., Maruthurkkara et al.], Remote Check uniquely combines technical CI checks with hearing performance tests").

This has now been added to the introduction:

Other applications for remote assessment of CI users have been developed (for example Philips et al., 2018, Maruthurkkara et al., 2022a, Maruthurkkara et al., 2022b). To the best of our knowledge, the present system is the first to incorporate PROM assessments.

3. Methods

• Clarify if consent included data publication (if applicable).

This is now clarified (it was).

• Inclusion/Exclusion:

o Specify why COMBI40X/C users were excluded (technical incompatibility?).

The COMBI40 implants are compatible with modern audio processors, however they can only use certain older sound coding strategies like CIS. Additionally, these implants have only eight channels compared to the twelve channels of modern implants.

For these reasons, they are frequently excluded from CI studies. Although they would be technically compatible with the present study, we also used this exclusion criterion for consistency with the modern literature.

As these implants were introduced in the early 1990s, there are likely few individuals worldwide still using them. No one was excluded on the basis of this criterion in this study. We have noted this in the methods.

o Clarify how "ability to operate a mobile device" was assessed (self-reported? screening test?).

It was self-declared proficiency. Now noted.

• MAUQ Translation: Briefly note if the translation was validated (or cite a reference for the method).

We note this in the limitation section:

The MAUQ was designed and published in the English language. To use it here, we performed an ad hoc translation of the items into Spanish using a forward and back-translation method. As such, it cannot be certain that the psychometric properties of this instrument (factor loadings, construct and criterion validity, and internal consistency) carry over into the Spanish language.

We have now moved this forward to the Methods section.

4. Results

• MAUQ Scores:

o The high scores (mean ~5.8/7) are compelling, but consider adding a comparison to benchmarks (e.g., "Scores exceed the ‘acceptable’ threshold of 4.5/7 in mHealth studies [cite]").

This is a fair point. We have added this to our Discussion under Summary of findings:

To the best of our knowledge, there are no established methods to categorize the results of the MAUQ (i.e. cutoff values for usability rated as poor, acceptable, good, etc.). Following Sanchez Roman et al. (2025), we suggest provisionally aligning MAUQ scores to the categories of the more general System Usability Scale (Bangor et al., 2009). This would yield lower cutoff values for the MAUQ of poor (2.7), OK (3.7), good (5.1), and excellent (6.0), and an overall acceptable usability threshold of 4.9.

o Address the ENT’s lower ratings in the Discussion (e.g., "ENTs may prioritize in-person exams for complex cases").

This is probably the reason. We have added it to the opening paragraph of the discussion.

• Cost Savings:

o The 48% "strongly agree" is impactful, but contextualize the one dissenting adult (e.g., "The outlier may reflect unique financial circumstances").

We have added the following to contextualized it:

We do not know the reasoning of the individual who strongly disagreed. It is reasonable to suppose that financial savings will be less for those who are not exposed to major costs (for example, those living close to the clinic or those who are retired and will do not need to take time off from work).

• Procedure Agreement:

o Highlight the 10% audiological disagreement rate as a limitation (e.g., "Audiological actions may require supplemental in-person checks").

We actually did address this in the discussion:

3 of 30 session pairs (10%) did not have agreement, all of which were for audiological follow-up actions. It is unclear why full agreement was not achieved with audiological actions. It may be the case that these types of actions require a relatively greater degree of in-person interaction.

5. Discussion

• Limitations:

o Sample Size: Acknowledge potential Type II errors (e.g., "Small n may underdetect differences in subgroups like pediatric vs. adult users").

Again, we have addressed this; given the sample size, we did not even attempt statistical inference, let alone try to detect significant difference between the pediatric and adult users:

Statistical inference from a cohort of this size is often accompanied by a degree of uncertainty and was, therefore, not attempted.

o Generalizability: Note if clinics were urban/rural or had high/low CI volumes.

We have included the type of hospital. To establish the volume of implants per year within a ratio, the German CI registry paper (PMID 38977469) has been used, which indicates the number of implants per year for each hospital, taking the average as a reference, it has been categorized as low, medium and high. This has been included in the Study Design section:

Both Donostia hospital and Virgen Macarena hospital are tertiary care centers. According to the distribution described in (T Stöver et al, 2024), the Donostia hospital falls within the medium/average cochlear implant volume range, whereas the Virgen Macarena corresponds to a large-volume category.

• Comparison to Prior Work:

o Emphasize how Remote Check differs (e.g., "Unlike hearDigits, Remote Check integrates PROMs and technical checks").

This has now been addressed in the introduction:

To the best of our knowledge, the present system is the first to incorporate PROM assessments.

• Clinical Implications:

o Recommend hybrid models (e.g., "Remote Check for routine checks; in-person for audiological fine-tuning").

This is an early phase feasibility study of a new technology. We are hesitant to advocate either a purely remote follow-up strategy or a hybrid strategy. This will surely vary by clinic. It would be inappropriate at this point to frame a one size fits all approach. With respect, we will decline to suggest an approach like this.

6. Language & Clarity

• Minor Edits:

o "Mobile-first principles" → "Mobile-optimized design" (more precise).

Changed, thanks.

o "Ad-hoc survey" → "Custom survey" (avoids informal tone).

Respectfully we disagree, “ad-hoc” is the common term used for an instrument that was designed for a specific study and which does not have external validation. The term “custom survey” might imply to the reader that we have validated it.

• Repetition: Some results (e.g., MAUQ scores) are duplicated; streamline.

We have gone through the manuscript thoroughly and ensured that there are no duplications in the results and discussion.

Reviewer#2

Introduction

The literature review is linked with the need for the study. However, an extensive literature review about the parameters focused on by the author in this manuscript is missing. Therefore, authors should include studies to provide better understanding.

At present the literature on apps that are integrated with CI remote care is a bit limited, as it is a very new topic. To the best of our knowledge we have included the relevant literature.

The author mentioned that “Follow-up sessions typically entail a visit by the user to a specialist at an audiology center. This can pose several challenges and inconveniences.” This needs supporting literature in specific to the challenges faced by the patient/caregivers apart from those mentioned in this paragraph.

We have tried to expand the supporting literature here.

I feel the author should mention and differentiate the mandatory visit to the hospital versus remote consultation, since this depends upon the duration of the implant use. Once a patient's device is switched on, they need more frequent visits, and it's essential to have face-to-face consultation, whereas after years of device use, their issues and consultation can be taken care of through remote control.

Agreed, we have now pointed out that the app is suited for users of all ages and degrees of CI experience.

In line 78-79, it is mentioned as “Ideally, routine follow-up sessions could take place remotely with the consultation and speech perception testing carried while the CI user was at home”. Can the author be specific about the speech perception testing possible to perform in the remote control mode?

We go into this in the Methods section. In the MED-EL app, speech perception tests are implemented as a three-digit test for speech-in-noise perception and the Ling-six perception test to assess access to speech sounds.

Ideally, it would be possible to implement a wider set of tests of speech perception, particularly monosyllabic words in quiet and sentences in noise, as are commonly used in clinical testing. There are a few hurdles to this. For one, the test material is generally proprietary, so the app would have to implement custom material which may not have the same psychometric properties as those used in the clinic. We have now noted this in the Discussion under a new section Speech perception tests.

In lines 82-85, it is mentioned as “MED-EL Remote Check is a web-based application which is designed to facilitate remote assessment of aided hearing performance and to collect patient reported outcome measures (PROMs) implemented as questionnaires.”,

The author should emphasize after how many months of the implant age or device uses, MEDEL remote check web based application can help gather the required information. Further, whether the questionnaire is targeted to the patient/caregivers/children/adults should be mentioned with a little more detail for better understanding of the readers.

We now note that “It is intended for users of all ages and with any degree of CI experience, (although younger children may require the assistance of adults).”

Method

In the method, it's mentioned as a multi-center, open-label, open-arm, prospective study. However, the data collection is done only from two centers.

We believe that two independent medical institutions (Donostia University Hospital in San Sebastian and Virgen Macarena University Hospital in Seville) meets the definition of a multi-center study.

In line 216-227, The details about the number of participants and other demographic details (unilateral/bimodal/bilateral, children, adult, pre-lingual/post-lingual) should be mentioned in the method chapter rather than in the results section.

We agree, we have moved this to the Methods.

It is not clear why authors consider different duration of experience across different professionals (ENT/Audiologist/SLT), which probably influences the findings.

We have found that in ENT papers where professionals are the research participants, it is common to give information about their duration of experience in the role. In this case, we have only three participants, so it is not really possible to address whether duration of experience has any influence on the results. Nevertheless we gave this information anyway.

In line 163-164, it is mentioned that “A forward and back-translation approach was used to produce a Spanish translation of the instrument”, is this validated on the Spanish individuals. If yes, authors can mention the same.

Reviewer 1 also asked this. At the time of the study, there was no validated Spanish version of the MAUQ (although one has since been produced). This is a limitation, as we note:

it cannot be certain that the psychometric properties of this instrument (factor loadings, construct and criterion validity, and internal consistency) carry over into the Spanish language.

Results

The information about the participants must be shifted in the method section.

Done, thanks.

What was the method adopted by the author to collect data on 29 CI users? Was this data sample appropriate to make inferences?

This was a preliminary study about experiences with an early-stage technology. The goal was to gather data about the challenges experienced by users that might be addressed through remote care (travel distance and times, frequency of visits, costs etc.), and to gather data about experiences with the Remote Check app itself. Recruitment was through convenience sampling. As the number of assessments per users was relatively large, we designed the study with a total number of participants of ~n=30 to avoid placing too much of a burden on both the users and on clinical scheduling. No inferential analysis was planned. The study was purely qualitative by design.

There are 29 CI users, but the data shown in the MAUQ response is of only 27 CI users. What about the remaining two participants?

Unfortunately, not all participants completed all assessments. The MAUQ was fully completed by only 20 individuals. This is a response rate of 71%. We have now added this in the limitations section:

Not all users completed all assessments, for example only 20 users fully completed the MAUQ (response rate 71%). We cannot know if there was a systematic bias such that non-responding users had different opinions.

In Table 2, only the mean is mentioned instead median and SD also needs to be added.

Agreed, we have added these values.

In user survey, again data is only about 28 CI users though the total participants are 29. Furthermore, there is a lot of variability observed among the participants.

One participant failed to complete the user survey (meaning a response rate of 96%). We would expect great variability among participants, as their frequency of visits, distance and travel times, costs, etc. will all vary based on individual life factors. This was not a controlled experiment where inter-subject variability is undesirable, instead it is a cross-sectional sample of individual experiences with their healthcare.

Under the professional survey, there are only 3 participants who participated, which limits the generalization of the findings.

Agreed, and noted as a major limitation.

The statistics used are only descriptive. There are no inferential statistics across the conditions, measures used to strengthen the finding.

Correct, we did not think inferential statistics would be suitable for a small and heterogenous cohort.

Discussion

The summary of the findings mentioned in the discussion section is actually a repetition of the information and is not essentially needed. It might not provide any additional understanding to the reader.

The discussion section written at present is, in general, rather specific to the findings of the present study or, the domain explored in the present study.

Thanks to the comments provided by both peer reviewers, we have now added much more contextualization to these results. Hopefully the revised discussion is more appropriate.

The author mentioned that “The results of the present study are broadly in accord with this, in that users demonstrated motivation to use the application and found it to be usable.”. How the present study findings are in line of the study done by Philips et al (2018). Did the author evaluate the motivation aspects during the remote check app application?

You are right, we totally over interpreted that. In both studies, a subjective usability scale was administered (MAUQ in ours and the System Usability Scale in Philips). In both cases, relatively high usabilit

---

## [Editor Report · Decision Letter 1]

7 Jan 2026

Evaluation of a mobile-based remote aftercare application in cochlear implant users

PONE-D-25-23826R1

Dear Dr. ALTUNA,

We’re pleased to inform you that your manuscript has been judged scientifically suitable for publication and will be formally accepted for publication once it meets all outstanding technical requirements.

Kind regards,

Rohit Ravi, Ph.D.

Academic Editor

PLOS One
---

## [Editor Report · Acceptance letter]

PONE-D-25-23826R1

PLOS One

Dear Dr. ALTUNA,

I'm pleased to inform you that your manuscript has been deemed suitable for publication in PLOS One. Congratulations! Your manuscript is now being handed over to our production team.

Kind regards,

on behalf of

Dr. Rohit Ravi

Academic Editor

PLOS One